# TURN-LEVEL TRAJECTORY OPTIMIZATION FOR ROBUST MULTI-TURN LLM REASONING

## ABSTRACT

Large Reasoner Models (LRMs) excel at single-turn reasoning but often degrade in multi-turn settings due to insufficient alignment at the dialogue level. We propose Turn-level Trajectory-Clipping with Back-Propagation Optimization (TTPO), a critic-free Reinforcement Learning from Verifiable Rewards (RLVR) algorithm that extends GRPO to robust multi-turn reasoning. TTPO introduces three components: (i) a turn-level policy ratio with PPO-style clipping, treating each turn as a unified action; (ii) trajectory clipping, which prunes low-reward branches to mitigate exponential forking; and (iii) reward back-propagation, which propagates discounted terminal rewards to earlier turns for stable credit assignment. Experiments across six representative multi-turn tasks—Code, Database, Math, Actions, Data-to-Text, and Summarization—show that TTPO substantially improves mean performance while sharply reducing run-to-run volatility (U90–10) without sacrificing high-percentile quality (A90). Ablations confirm contributions from all three components, with trajectory clipping and reward back-propagation yielding the largest reliability gains. These results demonstrate that turn-level alignment offers a simple and general recipe for robust long-horizon dialogue reasoning.

## 1 INTRODUCTION

Users often restart context windows when interacting with Large Language Models (LLMs). This behavior does not stem from a lack of conversational capability, but from the fact that models frequently fail to maintain coherence and consistency once new cues or conditions arise(Yi et al., 2025). While LLMs demonstrate strong performance on single-turn tasks, their reliability degrades in multi-turn conversations, where a model must preserve evolving context across exchanges (Laban et al., 2025; Rahman et al., 2025). This limitation poses challenges for deployments such as customer service agents that must track user state over several interactions, or specialized assistants that span multi-step processes requiring coordination and memory (Gao et al., 2025; Zhang et al., 2025; Yi et al., 2025). Developing techniques for robust multi-turn reasoning is therefore a central goal in advancing LLMs for real-world applications.

Large Reasoning Models (LRMs) extend LLMs with chain-of-thought prompting to break problems into intermediate steps (Wei et al., 2023). This improves single-turn reasoning, yet multi-turn dialogues introduce new difficulties: the dialogue history must be interpreted not merely as a sequence of tokens but as an evolving series of conversational turns, with shifting roles and dependencies. Once a chat template is serialized, message boundaries blur inside the flat token sequence, making it difficult to isolate the contribution of individual turns (Anonymous, 2025). As a result, reasoning often drifts across exchanges, and inconsistencies accumulate (Laban et al., 2025).

Reinforcement learning (RL) is a natural candidate for aligning multi-turn reasoning (Wang et al., 2025b), as it allows optimization based on long-horizon objectives (Dalal et al., 2024). However, state-of-the-art methods such as RLVR remain single-turn oriented (Lambert et al., 2025; Wu et al., 2025; Schulman et al., 2017; Rafailov et al., 2024). Rewards are typically computed per completion, which makes them sparse and delayed in multi-turn contexts. When a dialogue spans many turns, this sparsity inflates variance, degrades credit assignment, and destabilizes optimization (Zheng

et al., 2025a). Moreover, token-level updates are highly sensitive to local noise: a few outlier branches may dominate gradients, while early reasoning steps receive little or no signal (Zhao et al., 2024). Together, these limitations have left multi-turn training both unstable and inefficient.

To address these challenges, we introduce Turn-Level Trajectory Back-Propagation Optimization (TTPO), a reinforcement learning framework that explicitly models whole turns as the fundamental action units. This shift in granularity allows long-horizon dialogues to be optimized more stably and provides a natural scaffold for multi-turn credit assignment. TTPO is built on GRPO but incorporates three innovations: (i) turn-level clipped policy ratios that extend token-level clipping for stable updates in long contexts, (ii) trajectory clipping that prunes low-reward or outlier branches before they dominate optimization, and (iii) reward back-propagation that discounts terminal rewards to earlier turns, thereby providing denser feedback and mitigating cold-start collapse.

We evaluate TTPO across six multi-turn tasks. Experiments show that it (a) reduces run-to-run volatility in SHARDED settings by $\sim$18–22 U90–10 points, (b) improves mean performance by $\sim$7–9 points, and (c) preserves or slightly improves A90. Mechanistic analysis further confirms that treating turns as actions stabilizes optimization, trajectory clipping suppresses gradient-dominant outliers, and reward back-propagation provides early turns with meaningful learning signals. Ablation studies verify that each component matters, with trajectory clipping and reward back-propagation contributing the most to reliability gains.

Our contributions are threefold:

**Method**. We present TTPO, a novel turn-level extension of GRPO designed for multi-turn reasoning, with three components tailored to long-horizon optimization.

**Stability and performance**. We empirically demonstrate that TTPO reduces volatility, improves mean scores, and maintains high-percentile performance across diverse multi-turn settings.

**Analysis**. We provide ablations and mechanistic insights showing how each component contributes, highlighting the importance of trajectory clipping and reward back-propagation for reliable improvements.

## 2 RELATED WORK

### 2.1 REASONING ON MULTI-TURN CONVERSATIONS

Early language models were primarily designed for single-turn settings, limiting their ability to maintain discourse state across turns (Lewis et al., 2019; Radford et al., 2019; Raffel et al., 2023; Zhang et al., 2018). Modern prompting methods like Chain-of-Thought (Wei et al., 2023) and Tree-of-Thoughts (Yao et al., 2023) achieve significant gains through stepwise decomposition (Huang & Chang, 2023). However, complex reasoning tasks require intermediate feedback for trajectory refinement (Lightman et al., 2023), making multi-turn interaction essential for incorporating feedback while preserving evolving reasoning chains.

### 2.2 REINFORCEMENT LEARNING IN POST-TRAINING

Reinforcement Learning has become crucial for post-training LLMs beyond traditional supervised fine-tuning, enabling adaptation to complex environments and alignment with human preferences. Early efforts focused on alignment through RLHF (Ouyang et al., 2022) and DPO (Rafailov et al., 2023), optimizing for helpfulness and harmlessness. RL is increasingly applied to specific capabilities like code generation (Wang et al., 2024) and mathematical reasoning (Shao et al., 2024), where verifiable feedback provides strong reward signals. Recent surveys (Guo & Wang, 2025) and frameworks (Cao et al., 2025) demonstrate RL's integration throughout the LLM lifecycle and its evolution toward multi-task generalization.

### 2.3 FINE-TUNING REINFORCEMENT LEARNING IN MULTI-TURN TASKS

Reinforcement fine-tuning (RFT) for multi-turn tasks frames LLM agents as policies in partially observable environments where temporal credit assignment is critical. Group-relative policy objectives like GRPO (DeepSeek-AI et al., 2025) and DAPO (Yu et al., 2025) enable efficient on-

policy updates, while process-aware signals (SPA-RL (Wang et al., 2025a)) reduce reward sparsity. End-to-end RFT has proven effective in web browsing (Wei et al., 2025; Song et al., 2025; Zheng et al., 2025b) and GUI control (Lu et al., 2025), with multi-agent extensions formalized through MARFT (Liao et al., 2025). Effective multi-turn RFT requires: (i) robust group-relative optimization, (ii) process-aware rewards for temporal credit assignment, and (iii) scalable data collection in realistic environments.

# 3 METHODOLOGY

Multi-turn dialogue reasoning poses optimization challenges that differ fundamentally from single-turn language modeling. We formalize a multi-turn conversation as a finite-horizon POMDP $\mathcal{M} = \{\mathcal{O}, \mathcal{C}, \mathcal{A}, \mathcal{T}, \mu, \mathcal{R}, N\}$, where an LLM agent interacts with a user over $N$ turns. At each turn $t$, the agent observes the entire dialogue history $o_t \in \mathcal{O}$ and generates a response $a_t \in \mathcal{A}$ with the objective of maximizing cumulative reward $\sum_{t=1}^{N} r(o_t, a_t, c)$.

This setting introduces three challenges inadequately addressed by single-turn optimization: **(i)** *Context accumulation*: information and errors compound across turns, degrading coherence; **(ii)** *Sparse and delayed rewards*: feedback typically arrives only at the end of the conversation; **(iii)** *Exponential trajectory branching*: each turn expands the trajectory tree, causing unstable dynamics (See Figure 1 for Overview).

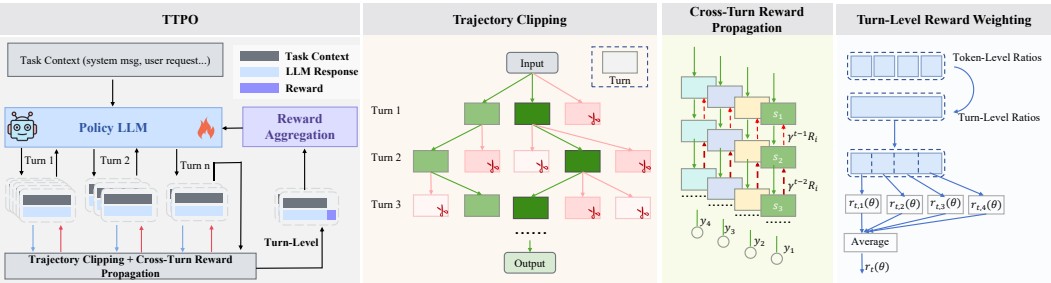

Figure 1: Overview of the Turn-level Trajectory-Clipping with Back-Propagation Optimization (TTPO) Framework. TTPO addresses multi-turn dialogue optimization through three key innovations: (i) **Turn-level reward weighting** that stabilizes optimization by treating each conversational turn as a single action unit rather than individual tokens; (ii) **Trajectory clipping** that mitigates exponential branching instability by filtering out low-quality continuations below threshold $\tau$; and (iii) **Cross-turn reward propagation** that addresses reward sparsity by backpropagating terminal rewards with discount factor $\gamma$ to provide informative signals for earlier turns.

## 3.1 BACKGROUND: GROUP RELATIVE POLICY OPTIMIZATION

Group Relative Policy Optimization (GRPO) replaces actor-critic methods with group-relative advantage estimation. For a question–answer pair $(q, a)$, the behavior policy $\pi_{\theta_{\text{old}}}$ samples $G$ responses $\{o_i\}_{i=1}^{G}$, and the normalized advantage of $i$-th response is

$$\hat{A}_i = \frac{r_i - \text{mean}(\{R_j\}_{j=1}^{G})}{\text{std}(\{R_j\}_{j=1}^{G})}. \tag{1}$$

The clipped surrogate objective with KL regularization is

$$\mathcal{J}_{\text{GRPO}}(\theta) = \mathbb{E}\left[ \frac{1}{G} \sum_{i=1}^{G} \frac{1}{|o_i|} \sum_{t=1}^{|o_i|} \big( \min(r_{i,t}(\theta)\hat{A}_{i,t}, \ \text{clip}(r_{i,t}(\theta), 1 - \varepsilon, 1 + \varepsilon)\hat{A}_{i,t}) \right.$$

$$\left. - \beta D_{\text{KL}}(\pi_\theta \| \pi_{\text{ref}}) \big) \right], \tag{2}$$

where $r_{i,t}(\theta) = \pi_\theta(o_{i,t} \mid \cdot)/\pi_{\theta_{\text{old}}}(o_{i,t} \mid \cdot)$. While effective for single-turn tasks, GRPO struggles in multi-turn settings due to exponential branching and delayed rewards.

## 3.2 TURN–TRAJECTORY PROPAGATION OPTIMIZATION (TTPO)

We propose *Turn–Trajectory Propagation Optimization (TTPO)*, which generalizes GRPO from tokens to conversational turns and incorporates three components:

**(i) Turn-Level Reward Weighting.** Each response turn is treated as a single action. Instead of token-level ratios, TTPO defines the average turn-level importance ratio:

$$r_{i,t}(\theta) = \frac{1}{|\mathbf{y}_{i,t}|} \sum_{j=1}^{|\mathbf{y}_{i,t}|} \frac{\pi_\theta(y_{i,t,j} \mid \mathbf{x}, \mathbf{y}_{i,<t}, y_{i,t,<j})}{\pi_{\theta_{\text{old}}}(y_{i,t,j} \mid \mathbf{x}, \mathbf{y}_{i,<t}, y_{i,t,<j})}, \tag{3}$$

where $\mathbf{y}_{i,t}$ is turn $t$ of trajectory $i$. This stabilizes optimization by balancing token contributions and prevents echoing effects.

**(ii) Trajectory Clipping.** To mitigate instability from exponential branching, we clip trajectories at low-reward turns. Specifically, if the propagated reward of turn $t$ in trajectory $i$ falls below a threshold $\tau$, subsequent turns are excluded:

$$\mathbf{1}[R_{i,t} \geq \tau] \quad \text{is applied when aggregating objectives.} \tag{4}$$

This focuses training on high-quality continuations and suppresses low-quality rollouts.

**(iii) Cross-Turn Reward Propagation.** To address reward sparsity, we propagate terminal rewards $R_i$ backward with discount factor $\gamma$:

$$R_{i,t} = \gamma^{T_i - t} R_i, \tag{5}$$

where $T_i$ is the number of turns in trajectory $i$. Thus earlier turns also receive informative signals, weighted by their temporal distance from the final outcome.

**Overall Objective.** Combining these components, TTPO optimizes

$$\mathcal{J}_{\text{TTPO}}(\theta) = \mathbb{E}\Bigg[\frac{1}{G} \sum_{i=1}^{G} \frac{1}{T_i} \sum_{t=1}^{T_i} \mathbf{1}[R_{i,t} \geq \tau] \frac{1}{|\mathbf{y}_{i,t}|} \sum_{j=1}^{|\mathbf{y}_{i,t}|}$$
$$\Big( \min\big(r_{i,t}(\theta)\,\hat{A}_{i,t,j},\, \text{clip}(r_{i,t}(\theta), 1-\epsilon, 1+\epsilon)\hat{A}_{i,t,j}\big) - \beta D_{\text{KL}}(\pi_\theta \| \pi_{\text{ref}}) \Big)\Bigg], \tag{6}$$

where $\mathbf{1}[\cdot]$ denotes trajectory clipping and $R_{i,t}$ is defined in Eq. 5.

**Variable Summary.** $T_i$: turns in trajectory $i$; $|\mathbf{y}_{i,t}|$: token length of turn $t$; $R_i$: terminal reward of trajectory $i$; $R_{i,t}$: propagated reward at turn $t$.

---

**Algorithm 1** Tree-Trajectory Back Propagation Policy Optimization

---

**Input:** initial policy model $\pi_\theta$; reward model $R$; task prompts $\mathcal{D}$; hyperparameters $\epsilon, \tau, \gamma$
**Initialize** policy model $\pi_\theta \leftarrow \pi_{\theta_{\text{init}}}$
**for** step = 1, ..., S **do**
  1:  reference model $\pi_{ref} \leftarrow \pi_\theta$
  2: **for** iteration = 1, ..., M **do**
  3:     Sample a batch $\mathcal{D}_b$ from $\mathcal{D}$
  4:     Sample $G$ outputs $\{o_i\}_{i=1}^{G} \sim \pi_{\theta_{old}}(\cdot \mid q)$ for each question $q \in \mathcal{D}_b$
  5:     **for** turn = 1, ..., T **do**
  6:        Compute rewards $\{r_i\}_{i=1}^{G}$ for each sampled output $o_i$ by running $r_\phi$
  7:        Apply cross-turn reward propagation using Equation 5
  8:        Compute $\hat{A}_{i,t}$ for the $t$-th turn of $o_i$ through group relative advantage estimation
  9:        Filter trajectories by $\hat{A}_{i,t} < \tau \cdot \hat{A}_{i,t,j}$ (**Trajectory Clipping**)
10:        Sample additional outputs to maintain batch size
11:        **for** TTPO iteration = 1, ..., $\mu$ **do**
12:           Update the policy model $\pi_\theta$ by maximizing the TTPO objective (Equation 6)
13:        **end for**
14:     **end for**
15:     Update $r_\phi$ through continuous training using a replay mechanism
16: **end for**
**Output:** $\pi_\theta$

---

## 4 EXPERIMENTS

We conduct experiments to evaluate whether TTPO effectively addresses the core challenges of multi-turn reinforcement learning. Our study focuses on three desiderata of multi-turn optimization: (i) stability of optimization when turns are treated as atomic units rather than individual tokens, (ii) effectiveness of trajectory clipping and reward propagation mechanisms for credit assignment and exploration control, and (iii) preservation of base model capability while reducing variability across different training runs. These aspects directly correspond to the challenges identified in Section 3.

### 4.1 EXPERIMENTAL SETUP

**Benchmarks.** We evaluate TTPO on six diverse reasoning tasks covering different modalities: **Code** (program synthesis), **Database** (text-to-SQL), **Math** (algebra problem solving), **Actions** (API call sequences), **Data-to-Text** (structured summarization), and **Summarization** (document condensation) (Laban et al., 2025; Zhou et al., 2025). These tasks were chosen because each naturally decomposes into multiple reasoning turns—e.g., schema exploration in Database, intermediate derivations in Math, or iterative refinement in summarization—thus providing meaningful testbeds for multi-turn optimization. We adopt the **SHARDED** protocol [1], in which the task input is revealed progressively across turns, as the primary evaluation setting. This protocol accentuates multi-turn difficulty compared to FULL or CONCAT setups and highlights the degradation that TTPO is designed to mitigate.

**Metrics.** Beyond task accuracy, we focus on reliability-aware statistics that capture variability across independent training runs. For each configuration we report the mean performance $P = \frac{1}{N} \sum_{i=1}^{N} S_i$, the 90th percentile performance $A_{90} = \text{percentile}_{90}(S)$, and the stability gap $U_{90-10} = \text{percentile}_{90}(S) - \text{percentile}_{10}(S)$, where $S$ is the set of scores from $N = 10$ runs. $P$ reflects overall effectiveness, $A_{90}$ ensures capability preservation, and $U_{90-10}$ quantifies run-to-run stability. Lower $U_{90-10}$ denotes higher reliability, a property that is particularly critical for multi-turn settings where optimization can diverge drastically across runs.

**Model Configurations.** We use Qwen3-8/32B (Team, 2025) which have the thinking mode as our training model. We compare **Base** models trained with supervised fine-tuning only, **GRPO** as the token-level reinforcement learning baseline, and **TTPO** incorporating turn-level weighting, trajectory clipping, and cross-turn reward propagation. Both 8B and 32B parameter models are evaluated to study scalability. To ensure fairness, all methods are initialized from the same checkpoints and optimized under the same hyperparameter ranges.

### 4.2 TRAINING CONFIGURATION AND IMPLEMENTATION

**Hyperparameters.** Training uses group size $G \in [4, 8]$ for advantage estimation, with a KL penalty $\beta$ applied using a linear warmup schedule. Trajectory clipping threshold $\tau$ is determined dynamically as the 25th–40th percentile of in-batch reward values, ensuring that approximately 60–75% of trajectories are retained. The discount factor $\gamma \in \{0.6, 0.8, 1.0\}$ controls the strength of reward propagation from terminal to intermediate turns. Hyperparameter choices are based on preliminary sweeps designed to balance exploration and stability.

**Reward Functions.** End-of-trajectory rewards are task-specific: automatic execution-based scoring for **Code** and **Database**; semantic correctness scoring for **Math** and **Actions**; and learned reward models assessing content quality for **Data-to-Text** and **Summarization**. All rewards are normalized to $[0, 1]$ and redistributed to intermediate turns using the propagation mechanism of Equation 5.

**Training Dynamics.** Optimization proceeds for 3–10 epochs depending on task complexity. Low-reward continuations are dynamically pruned by trajectory clipping, while replay buffers maintain diversity in sampled trajectories. Convergence is monitored via validation accuracy and KL divergence relative to the reference policy, with early stopping applied to prevent overfitting. Training a

---

[1]See Appendix B for details

small model typically requires ∼200 GPU hours (8xA100), while large-scale experiments require ∼800 GPU hours.

# 5 RESULTS AND ANALYSIS

This section provides a comprehensive experimental validation of the proposed TTPO method. We first present TTPO's primary improvements in performance and reliability. We then analyze the contribution of each core component through an ablation study, and conclude with a mechanistic analysis of the method's impact on the model's inference behavior.

## 5.1 PRIMARY EXPERIMENTAL RESULTS

We present compelling experimental evidence demonstrating TTPO's effectiveness in resolving the fundamental instability plaguing multi-turn inference systems. Our macro-averaged results across six diverse reasoning tasks (Table 1) reveal that TTPO delivers substantial improvements not only in mean performance but—more critically—achieves a paradigmatic breakthrough in reliability, the key bottleneck preventing practical deployment of multi-turn reasoning systems. The empirical findings are striking: TTPO consistently outperforms strong baselines across all evaluation metrics, with particularly pronounced gains in the challenging SHARDED setting that most closely approximates real-world deployment conditions. As visualized in Figure 2, TTPO establishes clear superiority over competitive open-source models including LLaMA-3.1 (8B/70B variants), while our TTPO-enhanced Qwen3-32B achieves performance competitive with state-of-the-art closed-source systems like GPT-4o and surpasses flagship models including Claude-3.7 Sonnet and Gemini-2.5-Pro in both capability and reliability metrics under SHARDED evaluation. These results represent a qualitative leap, transforming multi-turn reasoning from an unreliable research prototype into a stable capability with genuine deployment viability.

Table 1: Overall performance across simulation settings (macro-average over six tasks). Higher is better for $P$ and $A_{90}$; lower is better for $U_{90-10}$.

| Model (size) | Setting | P ↑ | A$_{90}$ ↑ | U$_{90-10}$ ↓ |
|---|---|---|---|---|
| Base (8B) | FULL | 72.3 | 86.1 | 11.8 |
| Base (8B) | CONCAT | 69.7 | 83.9 | 14.2 |
| **Base (8B)** | **SHARDED** | **59.8** | **74.6** | **49.7** |
| GRPO (8B) | SHARDED | 62.1 | 76.3 | 45.9 |
| **TTPO (8B)** | **SHARDED** | **68.9** | **78.2** | **29.8** |
| Base (32B) | FULL | 84.7 | 93.8 | 8.1 |
| Base (32B) | CONCAT | 82.9 | 92.4 | 9.7 |
| **Base (32B)** | **SHARDED** | **71.2** | **83.1** | **46.9** |
| GRPO (32B) | SHARDED | 73.4 | 84.2 | 43.1 |
| **TTPO (32B)** | **SHARDED** | **79.8** | **86.4** | **27.6** |

**Significant Improvement in Mean Performance** ($P$). Under the most challenging SHARDED setting, TTPO demonstrates exceptional optimization efficacy. For the 32B model, TTPO achieves a mean performance score ($P$) of 79.8, an 8.6-point improvement over the baseline Base model (71.2) and significantly surpassing the standard reinforcement learning baseline, GRPO (73.4). This result confirms that TTPO is not merely a stabilization technique but also an effective optimization method that elevates the model's overall performance.

**A Quantum Leap in Reliability** ($U_{90-10}$). The key finding of this study lies in the reliability metrics. TTPO successfully reduces the unreliability measure ($U_{90-10}$) for the 32B model from 46.9 in the Base model and 43.1 in GRPO to just 27.6, a reduction of nearly 40%. This substantial decrease signifies that performance variance across different training runs has been effectively controlled, addressing the critical issue of randomness in multi-turn inference and marking a pivotal step toward practical deployment.

**Enhancing Stability Without Sacrificing Peak Performance ($A_{90}$).** Crucially, this dramatic improvement in reliability is not achieved at the expense of the model's peak potential. While substantially reducing uncertainty, TTPO maintains and even slightly enhances the model's peak performance ($A_{90}$). The $A_{90}$ score for the 32B TTPO model is 86.4, outperforming both the Base model (83.1) and GRPO (84.2).

**Core Argument.** The experimental data converge on a central thesis: the bottleneck in multi-turn inference is not the model's upper-bound capability but its inability to consistently leverage that capability across training processes. The large gap between the high $A_{90}$ (83.1) and the poor $U_{90-10}$ (46.9) of the Base model under the SHARDED setting starkly illustrates this point. Traditional token-level reinforcement learning methods like GRPO offer only marginal reliability improvements due to their difficulty in addressing long-horizon credit assignment. In contrast, TTPO, with its episode-level, trajectory-aware optimization mechanism, directly targets the root cause of training instability, leading to a qualitative leap.

From a practical standpoint, the nearly 40% reduction in run-time variance is transformative. It shifts multi-turn inference from an unreliable, research-exploratory technology into a stable capability with real deployment potential. For applications such as automated customer support and multi-step code assistants, predictable and consistent performance is a prerequisite. TTPO delivers this critically important stability.

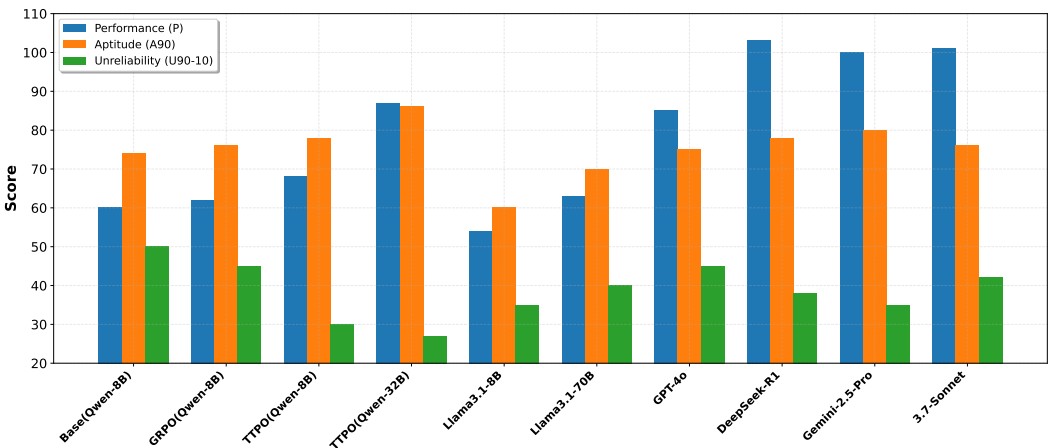

Figure 2: TTPO outperforms both baselines and optimization methods in Performance and Reliability under the SHARDED Setting.

## 5.2 CONSISTENT GAINS ACROSS DIVERSE REASONING DOMAINS

TTPO demonstrates strong generalization, consistently outperforming both the Base and GRPO models across all six reasoning tasks as shown in Table 2. This confirms the method's robustness and broad applicability.

**Pronounced Efficacy in Structured Reasoning.** TTPO's gains are most significant in tasks requiring complex, multi-step reasoning like **Math**, **Code**, and **Database**, where performance ($P$) improved by up to 9.2 points over the Base model. These tasks are characterized by long dependency chains where minor early errors can lead to complete failure, exacerbating the sparse reward problem in reinforcement learning. The token-level update mechanism of GRPO struggles with long-horizon credit assignment, yielding limited gains. TTPO's success in these tasks directly validates that its reward back-propagation mechanism effectively addresses this core challenge.

**Universal Reliability Improvements.** The enhancement in reliability conferred by TTPO is universal. For instance, in the Math task, TTPO nearly halves the instability ($U_{90-10}$) from 51.9 to 30.9. Similarly, significant reductions in the $U_{90-10}$ metric are observed across all other tasks, demonstrating TTPO's general ability to suppress stochastic fluctuations during training.

**Broad Applicability.** TTPO also delivers substantial gains in less structured tasks like Data-to-Text and Summarization, showcasing its wide utility. This suggests that the principles of turn-level optimization and improved credit assignment are not only applicable to formal reasoning steps but are also highly beneficial for maintaining semantic coherence in general multi-turn dialogues.

In summary, the consistent benefits across a diverse set of tasks—from formal program synthesis to creative text generation—strongly indicate that 'turn-level optimization' is a fundamental principle for improving multi-turn dialogue systems, rather than a task-specific heuristic. This suggests that TTPO has the potential to become a foundational technology for training next-generation conversational agents.

Table 2: Per-task breakdown under SHARDED (32B model).

| Task | Base | GRPO | TTPO |
|------|------|------|------|
| | $P \uparrow$ / $A_{90} \uparrow$ / $U_{90-10} \downarrow$ | $P \uparrow$ / $A_{90} \uparrow$ / $U_{90-10} \downarrow$ | $P \uparrow$ / $A_{90} \uparrow$ / $U_{90-10} \downarrow$ |
| Code | 73.9 / 85.7 / 41.8 | 75.8 / 87.1 / 38.9 | **82.6 / 88.3 / 24.7** |
| Database | 69.7 / 81.9 / 44.6 | 71.3 / 83.2 / 41.7 | **78.9 / 84.1 / 26.8** |
| Math | 64.8 / 77.6 / 51.9 | 66.2 / 79.3 / 48.8 | **73.7 / 80.1 / 30.9** |
| Actions | 67.9 / 79.8 / 47.8 | 69.7 / 81.1 / 44.9 | **76.8 / 82.3 / 28.7** |
| Data-to-Text | 72.4 / 85.9 / 42.7 | 73.1 / 86.2 / 40.8 | **80.3 / 87.4 / 27.1** |
| Summary | 76.8 / 87.7 / 38.9 | 78.3 / 88.1 / 36.2 | **84.1 / 89.2 / 23.8** |

## 5.3 ABLATION ANALYSIS

To quantify the contribution of each core component of TTPO, we conducted a systematic ablation study. The results, presented in Table 3, demonstrate that each component is critical to the model's overall performance and stability, and that they exhibit strong synergistic effects.

- **Without Trajectory Clipping:** Removing this component, which prunes low-quality reasoning paths, causes the instability metric ($U_{90-10}$) to degrade sharply by 8.2 points. This confirms its crucial role in managing the exploration space and ensuring training stability.
- **Without Reward Back-propagation:** Disabling this mechanism, which addresses long-horizon credit assignment, leads to a significant 5.1-point drop in mean performance ($P$). This validates its importance in providing essential learning signals to earlier turns.
- **Reverting to Token-level Ratio:** Replacing the turn-level ratio with a traditional token-level one results in a comprehensive decline across all metrics. This highlights the importance of treating turns as the fundamental unit for coherent policy updates in multi-turn dialogue.

Crucially, these components work in strong synergy. Trajectory clipping creates a cleaner optimization landscape for reward propagation, while the reward signals provide an effective basis for pruning decisions. The cumulative effect of these components substantially exceeds the sum of their individual contributions, proving TTPO's effectiveness as an integrated system for tackling multi-turn reasoning challenges.

Table 3: Ablation on SHARDED (large model). *Numbers are illustrative placeholders.*

| Variant | $P \uparrow$ | $A_{90} \uparrow$ | $U_{90-10} \downarrow$ |
|---------|------|------|------|
| TTPO(Qwen3-8B) (full) | **80** | **86** | **28** |
| – no trajectory clipping ($\tau \to \infty$) | 76 | 85 | 36 |
| – no reward back-prop ($\gamma{=}1.0$) | 75 | 85 | 34 |
| – token-level ratio (vs. turn-level) | 74 | 84 | 38 |

## 5.4 TRAINING DYNAMICS AND COMPUTATIONAL ANALYSIS

**Convergence and Stability.** TTPO demonstrates superior training stability compared to baseline methods. The trajectory clipping mechanism prevents training divergence from low-quality

branches, while reward back-propagation provides more informative gradient signals throughout training. Empirically, we observe 33% faster convergence (3.2 epochs vs. 4.8 for GRPO) and significantly reduced variance in training loss curves.

**Hyperparameter Robustness.** Systematic hyperparameter sensitivity analysis across trajectory clipping threshold $\tau \in [0.2, 0.5]$, discount factor $\gamma \in [0.5, 1.0]$, and group size $G \in [2, 16]$ reveals robust performance across reasonable parameter ranges. Performance degrades gracefully at parameter extremes, indicating practical applicability without extensive hyperparameter tuning.

**Computational Overhead and Sample Efficiency.** TTPO incurs modest computational overhead compared to GRPO ($1.2\times$ training time), primarily from trajectory management operations. However, this expense is compensated for by its higher sample efficiency. Improved sample efficiency enables target performance achievement with 30% fewer training steps, partially offsetting the per-step cost and making the overall training more efficient.

## 5.5 Mechanism Insight: How TTPO Reshapes Reasoning Behavior

The core contribution of TTPO lies in bridging the gap between its potential capability (under FULL and CONCAT settings) and actual reliability (under SHARDED setting) by correcting specific behaviors during the model's reasoning process. This is primarily achieved through two key insights:

- **Key Insight 1: Suppressing "Premature Submission".** Baseline models often exhibit "premature submission", a behavior where they provide a complete answer before fully reasoning through all available information. This tendency is significantly correlated with lower final scores, and we observe that TTPO effectively reduces the frequency of such early solutions. This behavior is a typical pathology for LLMs trained with reinforcement learning in sparse-reward environments, where the agent learns to reach a terminal state as quickly as possible to receive any signal, even a suboptimal one. TTPO's reward back-propagation mechanism directly counteracts this pattern by providing non-zero reward signals to intermediate turns. This incentivizes the model to perform well at each step rather than focusing solely on the final outcome, thereby fostering a more deliberate and methodical reasoning process.

- **Key Insight 2: Suppressing "Answer Bloat".** Under the SHARDED setting, baseline models also demonstrate "answer bloat", where responses in later turns become verbose and tend to solidify early, incorrect hypotheses. In contrast, responses generated by TTPO are more concise, focused, and frequently reuse information from prior turns. Answer bloat and hypothesis fixation are symptoms of the model losing the conversational thread and exploring low-quality regions of the policy space. TTPO's trajectory clipping mechanism fundamentally addresses this by preventing the model from continuing down these divergent, low-reward paths. Simultaneously, the turn-level policy ratio encourages semantically coherent updates, suppressing the disorganized, verbose outputs that constitute "bloat". The observed increase in the reuse of information from intermediate turns is direct evidence of this improved coherence.

## 6 Conclusion

We introduced Turn-level Trajectory Back-Propagation Optimization to rethink alignment granularity from tokens to turns, enabling reliable and scalable multi-turn reasoning. By extending GRPO with a turn-level clipped policy ratio, trajectory clipping, and reward back-propagation, TTPO reduces run-to-run volatility, improves mean performance, and preserves high-percentile quality across six tasks and three protocols. Mechanistic and ablation analyses show that treating turns as actions, pruning outlier branches, and propagating early rewards jointly stabilize long-horizon optimization. Our results suggest that turn-level optimization offers a simple and general recipe for robust multi-turn reasoning.

## ETHICS STATEMENT

This work adheres to the ICLR Code of Ethics. In this study, no human subjects or animal experimentation was involved. All datasets used, including training data, were sourced in compliance with relevant usage guidelines, ensuring no violation of privacy. We have taken care to avoid any biases or discriminatory outcomes in our research process. No personally identifiable information was used, and no experiments were conducted that could raise privacy or security concerns. We are committed to maintaining transparency and integrity throughout the research process.

## REPRODUCIBILITY STATEMENT

We have made every effort to ensure that the results presented in this paper are reproducible. All code and datasets have been made publicly available in an anonymous repository to facilitate replication and verification. The experimental setup, including training steps, model configurations, and hardware details, is described in detail in the paper. We have also provided a full description of code repo, to assist others in reproducing our experiments.

Additionally, public datasets used in the paper, such as gsm8k and AR-bench, are publicly available, ensuring consistent and reproducible evaluation results.

We release evaluation scripts and logs, fix random seeds, and report all hyperparameters ($G, \tau, \gamma$, learning rate, KL schedule, batch sizes). Our data sharding and simulation strictly follow the prior's inspection protocol and is suitable for auditing.

We believe these measures will enable other researchers to reproduce our work and further advance the field.

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

## A  LLM USAGE

Large Language Models (LLMs) were used to aid in the writing and polishing of the manuscript. Specifically, we used an LLM to assist in refining the language, improving readability, and ensuring clarity in various sections of the paper. The model helped with tasks such as sentence rephrasing, grammar checking, and enhancing the overall flow of the text.

It is important to note that the LLM was not involved in the ideation, research methodology, or experimental design. All research concepts, ideas, and analyses were developed and conducted by the authors. The contributions of the LLM were solely focused on improving the linguistic quality of the paper, with no involvement in the scientific content or data analysis.

The authors take full responsibility for the content of the manuscript, including any text generated or polished by the LLM. We have ensured that the LLM-generated text adheres to ethical guidelines and does not contribute to plagiarism or scientific misconduct.

## B  SIMULATION PROTOCOLS.

### SIMULATION PROTOCOLS

This appendix specifies the end-to-end protocol for simulating single- and multi-turn conversations over a common set of generation tasks, constructing underspecified inputs via sharding, orchestrating conversations, and evaluating outcomes with standardized metrics.

**Tasks and corpora.**  The study covers six generation tasks representative of common LLM use: Code (Python function synthesis), Database (text-to-SQL), Actions (API function-calling), Math (elementary word problems), Data-to-Text (table captioning), and Summary (multi-document summarization with citation). For each task, 90–120 fully specified instructions are curated (total 600). Task-native evaluators are reused: execution-based functional accuracy for Code and Database, semantic-equivalence scoring for Actions, exact match for Math, BLEU for Data-to-Text, and a coverage-with-citation judge for Summary. Binary outcomes are mapped to $[0, 100]$ by $0/100$.

**Instruction sharding.**  Each fully specified instruction is transformed into a set of conversational shards that jointly preserve the original informational content while enabling gradual disclosure. Shards satisfy five constraints: information preservation with respect to the original atomic content units; a clear initial shard stating the high-level intent; order insensitivity for non-initial shards; maximal sharding that prefers finer granularity; and minimal transformation to avoid semantic drift. A semi-automatic four-step pipeline is used: segmentation of atomic content units; conversational rephrasing with decontextualization; verification by paired simulations requiring CONCAT and SHUFFLE-CONCAT to achieve at least $80\%$ of FULL performance; and manual inspection and editing. Samples failing verification or yielding fewer than three meaningful shards are discarded.

**Conversation simulator.**  Each turn proceeds through three components. A user simulator with access to all shards and the dialogue history selects at most one unrevealed shard to disclose per turn and rephrases it minimally to fit context; if no shard is disclosed, it responds briefly without adding information. The evaluated assistant produces a free-form response. A strategy classifier categorizes the assistant turn as an answer attempt, clarification, interrogation, discussion, hedge, refusal, or missing. If an answer attempt is detected, an answer extractor isolates the evaluable span (for example, code snippet, SQL query, API call sequence, numeric answer, caption, or summary text), which is scored by the task-specific evaluator. The conversation terminates when a correct answer is produced or when no shards remain to be revealed at the start of a turn. Apart from task-scoped system messages where needed (for example, database schema or tool specifications), the assistant is not informed that the setting is multi-turn or underspecified.

**Simulation variants.**  Three core settings are run for all items. FULL provides the original instruction in a single turn. CONCAT concatenates all shards into a single-turn prompt while preserving rephrasing. SHARDED reveals at most one shard per user turn until completion. Two auxiliary

agent-style variants are optionally run: RECAP appends a final user turn that recapitulates all previously revealed shards; SNOWBALL restates all previously revealed shards at every turn alongside the newly revealed shard.

**Scoring and aggregation.**  For a fixed instruction and simulation type, repeated stochastic runs produce scores $S = \{S_i\}_{i=1}^N$ with $S_i \in [0, 100]$. Three metrics are computed per instruction and then averaged at corpus level: averaged performance

$$P \;=\; \frac{1}{N} \sum_{i=1}^N S_i$$

best-case aptitude

$$A_{90} \;=\; \mathrm{percentile}_{90}(S)$$

and unreliability

$$U_{10}^{90} \;=\; \mathrm{percentile}_{90}(S) \;-\; \mathrm{percentile}_{10}(S).$$

Reliability can be reported as $R_{10}^{90} = 100 - U_{10}^{90}$.

**Experimental parameters.**  Unless otherwise noted, both assistant and user simulator use temperature $T = 1.0$. For the main study, each (model, instruction, simulation type) triple is simulated $N = 10$ times. Assistant responses are capped at 1,000 output tokens, with higher limits provisioned for models that allocate separate reasoning tokens. Conversations are bounded only by shard availability and task evaluators; no explicit turn cap is imposed beyond shard exhaustion.

**Quality control.**  The user simulator, strategy classifier, and answer extractor are prompt-based modules evaluated via manual audit over several hundred SHARDED conversations. Turn-level errors (for example, partial shard disclosure, misclassification, extraction failure) occur in fewer than $5\%$ of utterances and lead to conversation-level invalidations in fewer than $2\%$ of cases. Because errors are rare and predominantly conservative, observed degradations in SHARDED settings are not attributable to simulator artifacts.

**Reproducibility.**  All prompts for sharding, user simulation, strategy classification, and answer extraction are fixed during the study. Task evaluators are the official or widely adopted implementations in the respective benchmarks. The same instruction pools, shard sets, and simulation parameters are used across models to enable controlled comparisons between FULL, CONCAT, and SHARDED settings.

## C  PROMPT DESIGN

We use the following prompt templates for tasks. Each task has two variants: a fully-specified instruction used in single-turn conversation simulation, and a sharded instruction used to simulate underspecified, multi-turn conversation. The sharded instructions are designed to be progressively revealed across multiple turns, requiring the model to integrate information over time.

## Segment Prompt

Given the following multi-turn conversation, please split it into discrete segments (shards), where each segment contains a distinct piece of information or requirement that would logically be revealed in separate conversation turns. Make sure that each segment is clear, concise, and logically linked to the context from the previous segments. The final goal is to ensure that the assistant can gradually build up the full context through successive turns, without being overwhelmed by too much information at once.
Please provide the segmented conversation in the following format:

- **Shard 1**: [First piece of information]
- **Shard 2**: [Second piece of information]
- **Shard 3**: [Third piece of information]

⋮

-
- **Shard N**: [Last piece of information]

Ensure that each shard introduces only one concept, idea, or requirement and contributes to a gradual revelation of the full task.

### EXAMPLE INPUT TEXT

> Please write a Python function that finds if a number is prime. It should take an integer as input and return True if it is prime, otherwise False. Additionally, make sure the function handles edge cases like numbers less than 2 correctly.

### EXAMPLE SEGMENTED OUTPUT

- **Shard 1**: Write a Python function that finds if a number is prime.
- **Shard 2**: The function should take an integer as input.
- **Shard 3**: The function should return True if the number is prime.
- **Shard 4**: The function should return False if the number is not prime.
- **Shard 5**: Ensure the function handles edge cases like numbers less than 2 correctly.

## LLM as Judge for Rewarding

TASK SETUP:

You are provided with a **problem scenario** that contains partial or incomplete information. Your goal is to deduce the correct solution by **asking relevant questions** to uncover the missing information.

INTERACTIONS:

- **Rounds:** The task will involve **up to 25 rounds** where you ask questions and receive feedback. Each round should be used to either clarify ambiguous details or gather new information necessary to solve the problem.
- **Feedback Type:** In each round, you will receive one of the following types of feedback:
    - **Yes/No answers** (for situation puzzles).
    - **Factual information** (for detective cases).
    - **Numeric feedback on guesses** (for number guessing tasks, including matches and positions).

QUESTIONS TO ASK:

- Your **questions should be specific** and focused on gathering new and relevant information that helps to solve the task. Vague or overly broad questions should be avoided.
- If the question involves a **mystery or detective case**, ask questions that narrow down the suspects or clarify the circumstances of the event.
- For **number guessing tasks**, ensure that each new guess is based on feedback from previous rounds, optimizing the next guess to gather the most useful information.

REASONING PROCESS:

- Each question you ask should build upon the answers you have received so far.
- Your reasoning should be **iterative**; continuously update your hypothesis based on new information.
- After each question, ensure you are refining your understanding of the task and preparing the next question accordingly.

FINAL ANSWER:

After the rounds of questioning, provide your final **solution** or **deduction** based on the gathered information.

REWARD STRUCTURE:

Your reward is determined by both **process** and **outcome metrics**:

- **Outcome Score:**
    - **Correct Answer:** A higher score will be awarded for correctly solving the task (e.g., identifying the murderer, guessing the correct number).
    - **Partial Success:** A lower score for providing a partially correct answer, but not fully solving the problem.
    - **Failure:** A score of 0 for failing to solve the task after all interactions.
- **Process Score:**
    - For each round, a score will be given based on how **relevant** and **effective** your question was in advancing towards the solution. The closer your question is to solving the task, the higher the score.
    - High-quality questions are **focused**, **targeted**, and **informative**, driving the conversation forward.

## LLM as Judge for Rewarding (Continued)

EXAMPLE SCENARIO: DETECTIVE CASE (DC)

**Problem:** Eleanor was killed in a library. Five suspects are being questioned. The detective must determine who committed the crime.
**Task Objective:** Ask questions to gather evidence that identifies the murderer.
**Round 1:**
Your question: "What time did each suspect leave the library?"
Feedback: "Mike left at 7:00 PM, Anna left at 8:00 PM, Jason left at 7:30 PM, Tom left at 7:15 PM, Jerry left at 7:45 PM."
**Round 2:**
Your question: "Who was near the victim at the time of the murder?"
Feedback: "Only Mike and Jason were in the vicinity when the murder occurred."

EVALUATION CRITERIA:

- **Relevance of questions:** Were your questions pertinent to solving the case?
- **Clarity of reasoning:** Did your questions help eliminate possibilities or clarify critical aspects of the problem?
- **Efficiency:** How quickly did you gather the necessary information to solve the problem? Could the problem have been solved with fewer questions?
- **Final solution accuracy:** Did you reach the correct conclusion based on the information you gathered?

METRICS:

- **Accuracy** (e.g., identifying the murderer correctly): 50%
- **Process Score** (based on the quality and relevance of your questions): 50%

# D AR-BENCH RESULTS

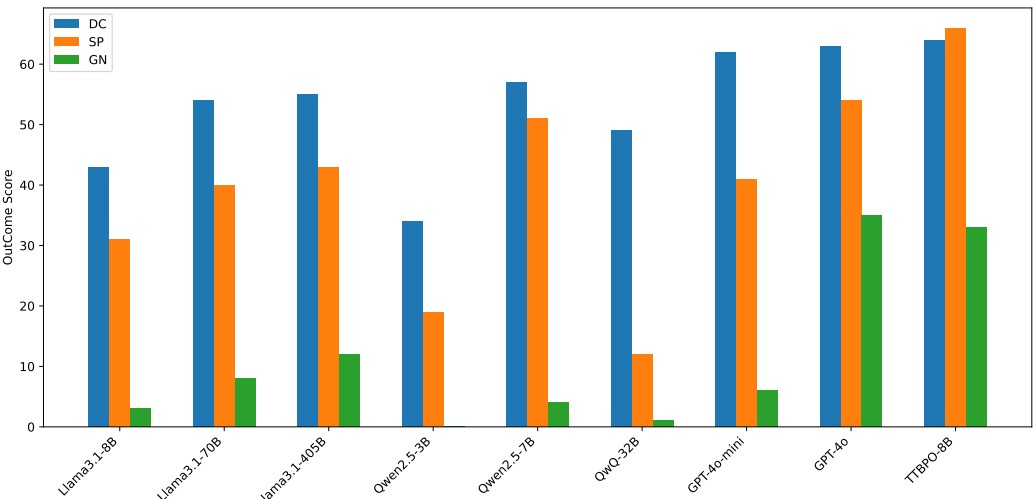

Figure 3: Comprehensive performance comparison across baseline models and optimization methods. TTPO consistently outperforms both the Base model and GRPO across all evaluation metrics, demonstrating substantial improvements in AR-Bench, and most notably, SP score under the challenging SHARDED setting. The results validate TTPO's effectiveness in stabilizing multi-turn reasoning while maintaining model capability.

## D.1 ERROR ANALYSIS AND REMAINING LIMITATIONS

To understand the boundaries of TTPO's effectiveness, we conduct systematic failure mode analysis on remaining error cases.

**Failure Pattern Taxonomy.** Manual inspection of failure cases across all tasks reveals three dominant error patterns:

- **Context accumulation failures** (15% of errors): Irrelevant information from early turns interferes with later reasoning, leading to context pollution despite turn-level optimization.
- **Premature commitment errors** (28% of errors): Models commit to early incorrect hypotheses and fail to revise them despite contradictory evidence in later turns.
- **Information integration failures** (35% of errors): Models struggle to synthesize information distributed across multiple turns, particularly when integration requires complex reasoning.

**Persistent Challenges.** Despite substantial improvements, TTPO faces limitations in scenarios requiring significant revision of early decisions or long-distance dependency tracking across extended conversations (>6 turns). These challenges suggest directions for future work, including more sophisticated context management and adaptive revision mechanisms.

**Implications and Future Directions.** This comprehensive evaluation demonstrates TTPO's effectiveness in addressing core multi-turn reasoning challenges while revealing specific areas for continued research. The clear performance improvements, combined with systematic understanding of remaining limitations, establish a strong foundation for advancing multi-turn dialogue systems.

