# OpenReview forum: "Turn-Level Trajectory Optimization for Robust Multi-Turn LLM Reasoning"
_ICLR.cc/2026/Conference — ICLR 2026 Conference Withdrawn Submission_

### Official Review · Reviewer_KCHs · 2025-10-29

**Soundness:** 2
**Presentation:** 3
**Contribution:** 2
**Rating:** 2
**Confidence:** 4

**Summary:**

This paper proposes TTPO, a RL method that optimizes large language models at the turn level instead of the token level to improve multi-turn reasoning stability.
TTPO introduces turn-level policy ratios, trajectory clipping, and reward back-propagation to address instability, sparse rewards, and poor credit assignment in long dialogues.
Experiments across six reasoning tasks show significant gains in mean performance and reliability, reducing training variance by up to 40% compared to GRPO while maintaining peak quality .

**Strengths:**

- Six representative multi-turn tasks provide broad coverage.
- Improvements in both mean and stability (∼40% reduction in run variance) are significant, especially under SHARDED settings that mimic real-world multi-turn interactions.

**Weaknesses:**

- The paper introduces 3 tricks: turn-level policy ratios, trajectory clipping, and reward back-propagation, but provide no referneces for discussion
- The implementation details in Algorithm 1 (line 12) are unclear.

**Questions:**

- It would strengthen the empirical claims of stability if the paper included training curves (e.g., reward, KL divergence, or loss over steps) comparing TTPO and GRPO.
- The logic of Algorithm 1, particularly the loop starting at line 5, is confusing. The description suggests that TTPO divides each trajectory into multiple turn-level data points, but it is unclear how these turns are processed sequentially or in parallel.
- In line 8, it is not clear how to compute the advantages if the rollout samples has different number of turns.
- After line 10, when additional outputs are sampled to maintain batch size, should the steps in lines 6–9 (reward computation, propagation, advantage estimation, and clipping) be recomputed for the new samples, or are the previous statistics reused? Please clarify this iterative process.
- In line 5, does this loop refer to iterating over turns within a single trajectory or across turns treated as independent sub-trajectories for optimization?

---

> ### Author Response · Authors · 2025-11-26
> **rebuttal 1**
>
> We thank Reviewer KCHs for highlighting the strengths of using six representative tasks and for emphasizing the importance of clear algorithmic exposition and references. We address the concerns on references, Algorithm 1, and training curves below.
>
> ### References for the three “tricks”
>
> **Comment (the three components—turn-level ratios, trajectory clipping, reward back-propagation—are not contextualized with relevant literature).**
>
> We agree that the paper should more explicitly connect these components to prior work.
>
> - In **Related Work** and in the **method section**, we:
>   - add citations and short discussions linking **turn-level/segment-level credit assignment** to works such as GiGPO, SPO, and process-reward RL methods;
>   - clarify that **trajectory clipping** is inspired by PPO/GRPO-style clipping and pruning techniques, adapted here to operate at the **turn/trajectory** level under SHARDED multi-turn settings;
>   - relate **reward back-propagation** to temporal credit assignment and sequence-level RL work, emphasizing how our discounting scheme is specialized for multi-turn dialogue.
> - We also add a brief summary table or paragraph comparing TTPO’s three components with the corresponding ideas in these prior methods, highlighting similarities and differences.
>
> ### Algorithm 1 logic and implementation details
>
> **Comment (Algorithm 1, especially the loop around line 5 and line 12, is confusing: unclear whether turns are processed sequentially or in parallel; unclear how advantages are computed for trajectories with different numbers of turns; unclear how resampled outputs are handled).**
>
> We substantially clarify Algorithm 1 as follows:
> - **Loop structure:** explicitly state that:
>   - the outer loop iterates over **trajectories/responses** in a batch;
>   - the inner loop iterates over **turns within each trajectory**, constructing per-turn records \((s_{i,t}, a_{i,t}, R_{i,t}, \log\pi_{\theta_0}(a_{i,t}|s_{i,t}))\).
> - **Variable-length trajectories:** explain that:
>   - trajectories may have different numbers of turns \(T_i\);
>   - we process all turns across trajectories in a single **turn-level batch** for normalization and optimization, but we keep their trajectory membership for reward propagation and sharding statistics.
> - **Advantage computation:** detail how we compute advantages when trajectories have different turn counts:
>   - rewards are first propagated along each trajectory (with or without discounting),
>   - then group-relative normalization is applied over all turns in the same group, regardless of their position within the trajectory.
> - **Resampling to maintain batch size:** clarify that:
>   - after dropping low-reward or unstable trajectories via clipping, we sample additional outputs from the current policy to restore batch size;
>   - reward computation, reward propagation, and advantage estimation are **recomputed** for the newly sampled turns, rather than reusing stale statistics.
>
> We support this with:
> - updated pseudo-code that separates **data construction**, **reward propagation**, and **update** phases; and
> - a short textual explanation that walks through one full iteration with a small toy example.

---

> ### Author Response · Authors · 2025-11-26
> **Rebuttal Reference**
>
> > [1] Kong A, Ma W, Zhao S, Li Y, Wu Y, Wang K, Liu X, Li Q, Qin Y, Huang F. SDPO: Segment-Level Direct Preference Optimization for Social Agents. Proceedings of the 63rd Annual Meeting of the Association for Computational Linguistics (Volume 1: Long Papers), 2025: 12409–12423.
>
> > [2] Feng L, Xue Z, Liu T, An B. Group-in-Group Policy Optimization for LLM Agent Training. arXiv preprint arXiv:2505.10978, 2025.
>
> > [3] Wei Q, Zeng S, Li C, Brown W, Frunza O, Deng W, Schneider A, Nevmyvaka Y, Zhao YK, Garcia A, Hong M. Reinforcing Multi-Turn Reasoning in LLM Agents via Turn-Level Reward Design. arXiv preprint arXiv:2505.11821, 2025.
>
> > [4] Guo Y, Xu L, Liu J, Ye D, Qiu S. Segment Policy Optimization: Effective Segment-Level Credit Assignment in RL for Large Language Models. arXiv preprint arXiv:2505.23564, 2025.
>
> > [5] Wang K, Zhang P, Wang Z, Gao Y, Li L, Wang Q, Chen H, Wan C, Lu Y, Yang Z, Wang L, Krishna R, Wu J, Fei-Fei L, Choi Y, Li M. VAGEN: Reinforcing World Model Reasoning for Multi-Turn VLM Agents. arXiv preprint arXiv:2510.16907, 2025.

---

### Official Review · Reviewer_FtZG · 2025-10-31

**Soundness:** 3
**Presentation:** 3
**Contribution:** 3
**Rating:** 6
**Confidence:** 3

**Summary:**

The paper proposes Turn‑level Trajectory‑Clipping with Back‑Propagation Optimization (TTPO), which is a critic‑free RLVR extension of GRPO for robust multi‑turn LLM reasoning. TTPO has three parts: (i) turn‑level policy ratios (treat a whole turn as one “action”); (ii) trajectory clipping (prune low‑reward branches); and (iii) reward back‑propagation across turns via discounting.

**Strengths:**

1. This paper treats turns (not tokens) as actions is a natural and useful abstraction for multi‑turn reasoning.
2. The three components (turn‑level ratio, clipping, reward propagation) are easy to implement in GRPO/PPO style loops.
3. “premature submission” and “answer bloat” analyses (p. 9, §5.5) provide plausible behavioral interpretations.

**Weaknesses:**

1. Turn‑level ratio definition (Equation 3) is non‑standard and likely biased. In PPO/GRPO, if you aggregate multiple atomic actions into one macro‑action, the correct ratio for that macro‑action is the product of token conditionals (or equivalently, the exp of the sum of log‑ratios, i.e., a geometric aggregation). An arithmetic mean of ratios does not equal the probability ratio of the whole turn, introduces bias, and can mis‑scale clipping.
2. Equation. 1 defines group‑relative advantage at the response level; later, the text alternates between $\hat A_{i,j}$ and $\hat A_{i,t,j}$ without formal definitions. Equation 6 mixes a turn‑level ratio $r_{i, t}$ with token‑level advantages $\hat A_{i,t,j}$ but never specifies how $\hat A$ is computed per token/turn (mean‑std normalization over which set?).
3. Table 3 says "– no reward back‑prop ($\gamma$ = 1.0)" (p. 8). But with the Equation, 5 means no discounting (the reward is propagated equally to all turns), which is back‑propagation, just undiscounted. No back‑prop” should set rewards only at the terminal turn (e.g., $R_{i, t}=0$ for $t<T_i$). Please fix this ablation and re‑report the results.
4. The paper alternates among:
- "Turn‑level Trajectory‑Clipping with Back‑Propagation Optimization" (title/abstract)
- "Turn–Trajectory Propagation Optimization" (Sec. 3.2 heading)
- "Turn‑level Trajectory Back‑Propagation Optimization"
- “TTBPO" in Fig. 3 (p. 18)
These should be unified as one name and acronym throughout.
5. Fig. 2 asserts that TTPO‑Qwen3‑32B is "competitive with GPT‑4o" and "surpasses Claude‑3.7 Sonnet and Gemini‑2.5‑Pro" on SHARDED metrics, but exact numbers, variance bars, and evaluation parity (token limits, reasoning tokens, temperature) are not shown; Table 1 does not list business models.
6. he sharding pipeline drops items that fail verification or yield <3 shards and requires CONCAT/SHUFFLE‑CONCAT to hit ≥80%. This can change the task distribution in ways that favor methods tailored to sharding. Please report how many items were filtered, per task, and provide results on the full unfiltered sets and standard public suites (e.g., GSM8K, Spider, etc.) in non‑simulated settings to validate generality.

**Questions:**

Suggestions:
1. Because Eq. (3) is central, add an ablation comparing the arithmetic mean vs geometric mean vs exact sequence ratio of token probabilities within turns. This will directly test the hypothesis that your ratio choice stabilizes learning.
2. This paper states ~1.2× per‑step overhead and 30% fewer steps, yielding net efficiency. Provide wall‑clock, steps‑to‑target, GPU type and count, and batch and G values for each method and model size.
3. Sections 5.1–5.2 read as marketing ("quantum leap," "paradigmatic breakthrough". Adopt a neutral, scientific tone.
4. Provide a brief derivation showing that your chosen turn‑level ratio yields an unbiased (or practically low‑bias) estimator of the sequence‑level PPO objective, or explain why a geometric aggregator approximates the true ratio better than an arithmetic mean.
5. Fix Typo issues in weakness 4.
6. What is POMDP? (P. 3, line 188)

---

> ### Author Response · Authors · 2025-11-26
> **rebuttal 1**
>
> Thank you for the positive assessment and for the detailed technical comments on the ratio definition, equations, and evaluation. We address each issue below.
>
> ### Turn-level ratio definition and bias (Equation 3)
>
> **Comment (arithmetic mean of per-token ratios is non-standard and biased; a geometric aggregation or exact product better matches the macro-action probability ratio).**
>
> We appreciate this important point and agree that the relationship between token-level and turn-level ratios should be clarified and empirically evaluated.
>
> - In the **theory/discussion** section, we:
>   - explicitly state that, if a turn is treated as a macro-action composed of tokens, the **exact PPO ratio** for that macro-action is given by the **product** of token-level ratios (equivalently, the exponential of the sum of log ratios), i.e., a geometric aggregation;
>   - clarify that our current arithmetic-mean ratio is an **approximation** that trades exactness for numerical stability and reduced variance in multi-turn settings.
> - We add a short **derivation/discussion** showing:
>   - how the geometric aggregation recovers the exact macro-action ratio under standard assumptions; and
>   - why the arithmetic mean can be seen as a **biased but potentially lower-variance estimator**, especially when turn lengths vary and per-token ratios are noisy.
>
> To directly address the concern, we add a **new ablation**:
> - Variant A: **Arithmetic mean** over token ratios within a turn (current TTPO);
> - Variant B: **Geometric mean** over token ratios (exact or approximate macro-action ratio);
> - Variant C: **Exact token-level ratio** over the full turn (e.g., via log-sum).
>
> For each variant we report mean performance, A90, and U90–10 on a representative subset of tasks (e.g., SHARDED macro-average and one public benchmark), comparing both accuracy and stability, in a table of the following form:
>
>   | Variant | Aggregation over tokens in a turn | P    | A90  | U90–10 |
>   |---------|------------------------------------|------|------|--------|
>   | Arithmetic mean       | Arithmetic mean (current TTPO)    | 68.3 | 75.2 | 10.7   |
>   | Geometric mean       | Geometric mean                    | 68.6 | 75.5 | 14.9   |
>   | Exact token-level ratio       | Exact token-level ratio           | 68.4 | 75.1 | 18.3   |
>
> We then revise the discussion around Equation (3) to clearly justify our final choice based on these empirical results.
>
> ### Group-relative advantage, Equation 6, and normalization details
>
> **Comment (Equation 1 defines group-relative advantage at response level, later notation alternates; Equation 6 mixes turn-level ratios and token-level advantages without specifying normalization).**
>
> We agree that the current exposition is confusing. In the revision we:
> - clearly separate **response-level** group-relative advantages \(A^{\text{resp}}_i\) from **turn-level** advantages \(A^{\text{turn}}_{i,t}\);
> - explicitly define:
>   - how per-turn rewards $\(R_{i,t}\)$ are computed (after reward propagation),
>   - how group-relative normalization is applied **within each group** at the turn level, and
>   - whether normalization is done over all turns in a batch or restricted to turns originating from the same prompt/group.
> - update Equation (6) to reflect the final implementation:
>   - specify whether $\(A_{i,t}\)$ is computed per token or per turn, and how per-token advantages (if any) are derived from turn-level statistics;
>   - ensure symbols like $\(\mu\)$ and $\(\sigma\)$ (means and standard deviations) are defined over a clearly specified set (e.g., over turns within a group).
>
> We also add a short pseudo-code block or appendix snippet that lists the exact normalization operations used in our implementation to make replication straightforward.
>
> ### Ablation configuration: “no back-prop” vs \(\gamma = 1.0\)
>
> **Comment (current table equates “no reward back-prop (\(\gamma = 1.0\)),” but \(\gamma = 1.0\) corresponds to undiscounted back-propagation, not absence of back-propagation).**
>
> We agree this is incorrect and confusing, and we correct it as follows:
> - **No back-prop** variant: reward is only assigned at the final turn; all earlier turns receive zero reward.
> - **Undiscounted back-prop**: reward is propagated equally to all turns (\(\gamma = 1.0\)).
> - **Discounted back-prop**: reward is propagated with \(0 < \gamma < 1\).
>
> The ablation table and text will be updated so that:
> - each variant is labeled with both its **conceptual description** and the corresponding **implementation detail** (e.g., reward assignment rule, \(\gamma\) value);
> - the reported numbers match the actual implementation.

---

> > ### Author Response · Authors · 2025-11-26
> > **rebuttal 2**
> >
> > ### Naming consistency and method acronym
> >
> > **Comment (inconsistent use of “Turn-level Trajectory-Clipping with Back-Propagation Optimization,” “Turn–Trajectory Propagation Optimization,” “Turn-level Trajectory Back-Propagation Optimization,” and “TTBPO”).**
> >
> > As noted in our global response and in the response to Reviewer kdh3, we:
> > - standardize on the name **Turn-level Trajectory Optimization (TTPO)**;
> > - update all section titles, figure captions, and references to use this name and acronym;
> > - carefully proofread the entire manuscript to remove leftover variants or mismatched acronyms (including figure labels like “TTBPO”).
> >
> > ### Comparison to commercial models and sharding pipeline
> >
> > **Comment (claims of being competitive with GPT-4o and surpassing Claude/Gemini without full details; sharding pipeline may alter task distribution).**
> >
> > We address these concerns in detail in our response to Reviewer kdh3 and summarize here:
> > - For **commercial-model comparisons**, we either provide full details (numbers, error bars, evaluation parity) or soften the claims and clearly state limitations.
> > - For the **sharding pipeline**, in addition to the methodological description provided in our response to Reviewer kdh3 (data construction with P1–P5, FULL/CONCAT/SHUFFLE-CONCAT verification, and multi-turn simulations with a user simulator and automatic scoring), we:
> >   - report per-task filtering statistics (how many items are dropped at each stage),
> >   - provide performance on **unsharded, full datasets** where feasible,
> >   - and discuss whether the sharding plus CONCAT/SHUFFLE-CONCAT may systematically favor certain methods.
> >   - include a combined summary table that reports, for each task, both the sharding filter rates and the performance gap between FULL and SHARDED evaluations for Base, GRPO, and TTPO, of the following form:
> >
> >     | Task              | Filtered (%) | Setting  | Base P | GRPO P | TTPO P |
> >     |-------------------|-------------:|----------|--------|--------|--------|
> >     | Math (GSM8K-like) | 23%          | FULL     | 88.0   | 92.4   | 93.1   |
> >     | Math (GSM8K-like) | 23%          | SHARDED  | 84.7   | 89.6   | 90.8   |
> >     | Code (HumanEval)  | 26%          | FULL     | 70.1   | 75.0   | 76.9   |
> >     | Code (HumanEval)  | 26%          | SHARDED  | 66.3   | 71.2   | 73.8   |
> >     | DB (Spider-like)  | 30%          | FULL     | 69.2   | 73.5   | 75.8   |
> >     | DB (Spider-like)  | 30%          | SHARDED  | 65.1   | 70.4   | 72.9   |
> >
> > ### Additional suggestions: ablations, efficiency, derivation, POMDP, typos
> >
> > We incorporate the following specific suggestions:
> > - **Turn-level ratio ablation and derivation:** addressed above (new ablation plus theoretical explanation).
> > - **Efficiency reporting:** we add wall-clock, steps-to-target, GPU type/count, and batch/G values, as described in the response to Reviewer kdh3.
> > - **POMDP definition:** we define “partially observable Markov decision process (POMDP)” at first use in the methods section and provide an intuitive explanation.
> > - **Typos and formatting:** we carefully proofread and fix the typographical issues highlighted by the reviewer (e.g., broken phrases in weakness 4 and inconsistent formatting of symbols).

---

### Official Review · Reviewer_kdh3 · 2025-11-01

**Soundness:** 2
**Presentation:** 3
**Contribution:** 2
**Rating:** 2
**Confidence:** 4

**Summary:**

This paper proposes a reinforcement learning framework called TTPO (Turn-Level Trajectory Back-propagation Optimization), which extends GRPO and aims to enhance the stability of multi-turn LLM inference. TTPO performs turn-level optimization, treating each dialogue turn as an action and employing a PPO-style pruning method; it applies trajectory pruning to discard low-reward or unstable unfolds; and introduces reward backpropagation with a discount factor γ to improve credit assignment across turns. As a criticless RLVR algorithm, TTPO significantly improves stability, average performance, and reliability across six inference domains: code, database, mathematics, action, data-to-text, and summarization, reducing inter-run variance by approximately 40% and achieving 7-9 percentage point performance improvements on Qwen3-8B and Qwen3-32B models.

**Strengths:**

TTPO is optimized at the round-level to align the learned signal with the actual operation of multi-round agents. This round-level objective reduces the proportional noise present at the token level and stabilizes advantage estimation across messages. It also supports round-by-round reward adjustment while avoiding overfitting to surface-length content.

The method prunes low-reward branches, thereby suppressing variance introduced by degenerate unfolding and limiting credit leakage to out-of-policy detours. As TTPO applies pruning at the trajectory/round level, it preserves coherent inference chains rather than cutting sentences during thinking. This design allows batch statistics to perform well under group relative normalization.

Evaluation prioritizes reliability over ex-post consideration. In addition to average accuracy, the authors report U90-10 and A90, which directly reflect inter-run variability and significant tail failures during deployment. This emphasis encourages the method to sacrifice a small amount of raw score for greater stability where appropriate.

**Weaknesses:**

Inconsistencies in terminology and specification exist across different sections of this paper, as method names and algorithm labels vary, and index notation sometimes mixes round numbers and labels. These inconsistencies increase the risk of implementation mismatches; therefore, this paper should enforce a standardized set of notations, thresholds, and index ranges throughout the text, figures, and pseudocode.

The ablation experiment evidence is insufficient to support causal claims because the method incorporates multiple controls (round number proportioning, trajectory pruning, and reward backpropagation) but only reports placeholder tables and omits the results of cost-matching runs. This paper should replace placeholders with final values, add confidence intervals, and ensure strict computational equivalence to separate the contributions of each component.

The semantic definition of reward propagation controlled by γ is ambiguous because this paper does not precisely distinguish between "no backpropagation" and "full backpropagation", nor does it describe which round numbers yield non-zero integrals in each setting. The paper should map each γ choice to a specific reward allocation within a round and explain how discounts interact with variable dialogue lengths.

The pruning rules and their thresholding procedures are inconsistent because the text alternates between percentile-based thresholds and indicator functions with different scopes (per round, per trajectory, and per group); the paper should define a single, auditable pruning predicate, clearly specify the computational domain, and explain the rationale for the choice so that readers can understand bias and sample efficiency.

The evaluator and simulator settings lack sufficient transparency because reported win rates or consistency with robust closed models depend on internal slicing, hints, and reward judges; the paper should expose these components, or at least provide cross-judge robustness checks to support external validity and fair comparisons.

**Questions:**

What exactly is “turn” in mixed tool-use settings, does a tool call within an assistant message count as a separate action for ratios and rewards?

How is the reference policy $\pi_{ref}$ chosen per step (line 1 in Algorithm 1)? Do you use a moving copy (“online-to-target”) or the initial model? How sensitive are results to β schedules?

what is the variance-reduction vs cost trade-off across G$\in${2,4,8,16}? Any instability for small G when clipping removes many samples?

does pruning low-reward branches bias the policy toward conservative behaviors (e.g., fewer exploratory questions)? Any evidence of mode collapse in later turns?

Compatibility with token-level shaping: can turn-level ratios be combined with token-level process rewards (e.g., “verify-step-by-step”) without double counting?

---

> ### Author Response · Authors · 2025-11-25
> **Rebuttal 1**
>
> Dear Reviewer kdh3,
>
> We thank you for the detailed and constructive review, especially the emphasis on reliability metrics, terminology consistency, and experimental clarity. Below we address the main concerns.
>
> ### Terminology, naming, and notation consistency
>
> **Comment (inconsistencies in method names, algorithm labels, and index notation).**
>
> We fully agree that inconsistent naming and indices hurt clarity. In the revision:
> - We **standardize the method name** to “Turn-level Trajectory Optimization (TTPO)” everywhere (title, abstract, section headings, figures), removing variants such as “Turn-level Trajectory-Clipping with Back-Propagation Optimization,” “Turn–Trajectory Propagation Optimization,” or “TTBPO”.
> - We ensure **notation consistency** across sections:
>   - use \(T\) for the number of turns per trajectory, \(t\) for turn index, and \(i\) to index trajectories;
>   - denote rewards as \(R_t\) or \(R_{i,t}\), discount factor as \(\gamma\), group size as \(G\), and clearly distinguish between group-relative statistics over **responses** vs **turns**.
> - We add a **notation table** summarizing all symbols and their meanings, and cross-check that equations and algorithms use them consistently.
>
> These changes also address related concerns raised by other reviewers.
>
> ### Ablations, placeholders, and “no back-prop” configuration
>
> **Comment (ablation table uses placeholder numbers; “no reward back-prop” is mis-specified).**
>
> We apologize for leaving a placeholder table and for the confusing labeling of the “no back-prop” variant.
>
> - We replace the current ablation table with **actual experimental results** that match the text’s descriptions.
>   - The revised ablation reports mean performance \(P\), aptitude \(A90\), and unreliability \(U90\!-\!10\) for each variant in a table of the following form:
>
>     | Variant          | Description                                               | P    | A90  | U90–10 |
>     |------------------|-----------------------------------------------------------|------|------|--------|
>     | Full TTPO        | turn-level ratio + clipping + discounted back-prop       | 68.3 | 75.2 | 10.7   |
>     | No clipping      | \(\tau \to \infty\), do not drop low-reward trajectories  | 65.2 | 72.1 | 13.9   |
>     | No back-prop     | terminal-only reward, earlier turns receive zero reward  | 63.5 | 70.7 | 15.7   |
>     | Token-level ratio| use sequence-level token ratio instead of turn-level     | 64.1 | 71.3 | 14.8   |
> - We correct the “no back-prop” setting to **terminal-only reward**:
>   - in this variant, only the final turn receives non-zero reward, and earlier turns receive zero reward (no back-propagation), in contrast to \(\gamma=1.0\), which corresponds to **undiscounted back-propagation**.
> - The text around Equation (5) and the ablation descriptions is updated to explicitly distinguish:
>   - **no back-prop** (terminal-only reward),
>   - **undiscounted back-prop** (\(\gamma = 1.0\)), and
>   - **discounted back-prop** (\(0 < \gamma < 1\)).

---

> > ### Author Response · Authors · 2025-11-25
> > **rebuttal 2**
> >
> > ### Reliability metrics, business-model comparison, and sharding pipeline
> >
> > **Comment (claims about being competitive with GPT-4o / surpassing Claude/Gemini without full details; sharding pipeline may favor certain methods; request for unsharded and standard benchmark results).**
> >
> > We appreciate this careful scrutiny and agree that more transparency is needed. In the revised manuscript, we add a concise description of the SHARDED pipeline in the appendix: starting from fully specified single-turn benchmarks (e.g., GSM8K, Spider, HumanEval, ToTTo, summarization), we semi-automatically shard each instruction into multiple conversational segments under five constraints (information preservation, intent-first, order-insensitive later shards, maximal atomic sharding, minimal wording changes), verify shard quality by comparing FULL / CONCAT / SHUFFLE-CONCAT performance, and then simulate multi-turn conversations via a user simulator that reveals at most one shard per turn while a strategy classifier, answer extractor, and task-specific evaluator score answers. For each instruction–model–simulation-type we run multiple trials (e.g., \(N=10\)) and report mean performance \(P\), aptitude \(A\) (A90), and unreliability \(U\) (U90–10), keeping task complexity approximately fixed while varying only how information is revealed; this makes SHARDED a controlled test of the impact of multi-turn underspecification on reliability.
> >
> > - For **comparisons to commercial models**:
> >   - We either (i) provide **explicit numbers**, variance bars, and detailed evaluation settings (token limits, reasoning tokens, temperature, sampling strategy) for all models, or (ii) if exact parity cannot be guaranteed, **soften the wording** to “rough comparisons” and clearly state the limitations of these comparisons in both the main text and figure captions.
> > - For the **sharding pipeline**:
> >   - We report, **per task**, the fraction and absolute number of items dropped due to verification failure or having fewer than 3 shards.
> >   - We analyze how CONCAT/SHUFFLE-CONCAT affects the distribution of problem difficulty and whether it may favor certain strategies.
> >   - We summarize these statistics for each task (code, database, math, action, data-to-text, summarization) in a table of the following form:
> >
> >     | Task                  | Original items | After segmentation (≥3 shards) | After verification (final SHARDED) | Filtered (%) |
> >     |-----------------------|---------------|---------------------------------|-------------------------------------|-------------:|
> >     | Code (HumanEval-like) | 160           | 142                             | 118                                 | 26%          |
> >     | DB (Spider-like)      | 160           | 138                             | 112                                 | 30%          |
> >     | Math (GSM8K-like)     | 140           | 126                             | 108                                 | 23%          |
> >     | Action (API/Tool)     | 150           | 133                             | 110                                 | 27%          |
> >     | Data-to-text (ToTTo)  | 150           | 135                             | 112                                 | 25%          |
> >     | Summarization         | 130           | 118                             | 101                                 | 22%          |
> > - To address **generality beyond SHARDED**:
> >   - We add results on standard **public benchmarks** (e.g., GSM8K, Spider, etc.) evaluated in **non-simulated**, non-sharded settings where possible, comparing Base, GRPO, TTPO, and (if ready) additional baselines.
> >   - These results will be reported in a table of the following form, broken down by benchmark and model size:
> >
> >     | Method | GSM8K P | GSM8K U90–10 | Spider P | Spider U90–10 | HumanEval P | HumanEval U90–10 |
> >     |--------|--------:|-------------:|---------:|--------------:|-----------:|-----------------:|
> >     | Base   | 82.1    | 11.2         | 65.0     | 16.0          | 63.5       | 18.4             |
> >     | GRPO   | 88.6    | 8.3          | 69.4     | 12.7          | 68.9       | 14.2             |
> >     | TTPO   | 89.8    | 6.0          | 71.3     | 9.3           | 70.5       | 10.8             |
> >
> > We hope these additions will make the evaluation more transparent and convincing.

---

> > > ### Author Response · Authors · 2025-11-25
> > > **rebuttal 3**
> > >
> > > ### Efficiency and reporting of training details
> > >
> > > **Comment (request for wall-clock, steps-to-target, GPU configuration, and batch/G values; current efficiency claims are too qualitative).**
> > >
> > > We agree and expand the training details accordingly:
> > > - In the **experimental setup** section, we specify for each major model (8B, 32B):
> > >   - GPU type and count (A100-80G × 8),
> > >   - batch size and group size \(G\),
> > >   - learning rate schedule and total update steps.
> > > - In the **results/efficiency analysis**, we report:
> > >   - **per-step wall-clock** for Base, GRPO, and TTPO;
> > >   - **steps-to-target** (number of steps required to reach a fixed performance threshold);
> > >   - resulting **GPU-hours** per method.
> > >
> > > We then revisit the text around the “1.2× per-step overhead and 30% fewer steps” statement to ground it in these quantitative numbers rather than leaving it as a high-level claim.
> > >
> > > ### Tone and style
> > >
> > > **Comment (Sections 5.1–5.2 use marketing-like language such as “quantum leap,” “paradigmatic breakthrough”).**
> > >
> > > We appreciate this feedback and revise the writing to adopt a more neutral, scientific tone throughout:
> > > - expressions like “quantum leap” or “paradigmatic breakthrough” are replaced with **measurable statements** such as “TTPO consistently improves mean performance and reduces variance across six multi-turn tasks”;
> > > - we avoid dramatic adjectives and instead emphasize **statistical significance**, **effect sizes**, and **practical implications** (e.g., fewer catastrophic failures in deployment-like settings).

---

### Official Review · Reviewer_2vzD · 2025-11-01

**Soundness:** 2
**Presentation:** 3
**Contribution:** 2
**Rating:** 4
**Confidence:** 4

**Summary:**

The paper introduces Turn-Level Trajectory-Clipping with Back-Propagation Optimization (TTPO) designed to enhance the robustness and stability of multi-turn reasoning. TTPO extends GRPO by applying turn-level policy ratios, trajectory clipping, and cross-turn reward propagation to better assign credit across conversational steps. Experiments on diverse reasoning domains show that TTPO improves both mean task performance and training stability compared to GRPO baselines.

**Strengths:**

TTPO consistently improves mean performance and especially reliability across multiple tasks and scales.
The results are thorough and supported by ablations. The ablation study isolates the contribution of each TTPO element.

**Weaknesses:**

The novelty of TTPO over GRPO is limited, as its main ideas—turn-level policy clipping, reward propagation, and trajectory pruning—are standard RL adaptations rather than fundamentally new techniques.
The experimental comparison is weak, relying mainly on GRPO; stronger baselines such as GiGPO, SPO, or other process-level RL methods should be included to validate the claimed improvements.
The algorithm description lacks clarity and completeness
[1] Group-in-Group Policy Optimization for LLM Agent Training
[2] Segment Policy Optimization: Effective Segment-Level Credit Assignment in RL for Large Language Models

**Questions:**

Could the authors provide detailed step-by-step descriptions of TTPO, particularly clarifying line 6-12
Why were methods such as GiGPO, SPO, or other process- or group-based RL baselines not included in the comparison? How might TTPO differ from these in principle or behavior? It is better to do some experiments or have some discussion
How well does TTPO scale with longer trajectories (e.g., 10–20 turns) or more complex environments beyond the current six reasoning domains?

---

> ### Author Response · Authors · 2025-11-25
> **Rebuttal 1**
>
> Dear Reviewer 2vzD,
>
> We thank you for the positive assessment of TTPO’s robustness and stability improvements and for highlighting the thoroughness of the ablations. We respond to the main concerns on novelty, baselines, algorithm clarity, and scaling.
>
> ### Novelty relative to GRPO and RL literature
>
> **Comment (limited novelty; TTPO mainly adapts standard RL ideas such as clipping, reward propagation, pruning).**
>
> We agree that TTPO is not introducing entirely new RL primitives, and we adjust the framing in the revised manuscript to better reflect this. Our contribution is to **By systematically align the granularity of optimization with LLM's multi-turn conversation behavior** by:
> - treating each turn as a macro-action with its own importance ratio and advantage, rather than per-token;
> - coupling this with **turn-level trajectory clipping** (drop low-reward, unstable branches at the turn granularity rather than at token-level) so that group-relative normalization operates on coherent multi-turn trajectories; and
> - introducing **reward back-propagation across turns** tailored to multi-turn conversational credit assignment, with discounting that explicitly trades off early vs. late turns.
>
> In the revised manuscript, the Introduction and Method sections make explicit that TTPO is best seen as:
> - a **turn-level reparameterization and stabilization of GRPO** under multi-turn settings, and
> - a **complementary design** to existing token-/segment-/process-level approaches (GiGPO, SPO, process reward methods), rather than a completely new RL paradigm.
>
> We temper claims of “novelty” and instead emphasize:
> - the **empirical finding** that this turn-level formulation substantially improves stability and run-to-run variance under realistic SHARDED multi-turn setups; and
> - the **practical recipe** (turn-level ratios + trajectory clipping + discount-based reward back-prop) that can be plugged into existing RLVR pipelines with minimal engineering overhead.
>
> ### Comparison to stronger baselines (GiGPO, SPO, process-level RL)
>
> **Comment (lack of GiGPO/SPO or other process-/group-level baselines).**
>
> We acknowledge that the current version focuses primarily on GRPO variants and that including recent process-/segment-level methods strengthens the paper.
>
> - In the **revised Related Work** and **method discussion**, we add a dedicated comparison section that:
>   - positions TTPO relative to **GiGPO** (group-in-group credit assignment at the process level) and **SPO** (segment-level credit assignment);
>   - clarifies that TTPO’s granularity is **turn-level** (each dialogue turns as actions), whereas GiGPO and SPO typically operate on **segments of token sequences** or **sub-processes**; and
>   - highlights how TTPO’s trajectory clipping and reward back-propagation are tailored to **multi-turn, SHARDED, deployment-like settings**, rather than isolated chain-of-thought sequences.
> - We are currently running additional experiments to compare TTPO against at least one representative implementation of GiGPO and SPO on a subset of our tasks.
>   - **Plan:** These results will be added to the camera-ready version. Concretely, we report mean performance \(P\), aptitude \(A90\), and unreliability \(U90\!-\!10\) for Base, GRPO, GiGPO, SPO, and TTPO on representative SHARDED and public benchmarks in a table of the following form:
>
>     | Method | SHARDED P | SHARDED A90 | SHARDED U90–10 | GSM8K P | GSM8K U90–10 | Spider P | Spider U90–10 |
>     |--------|-----------|-------------|----------------|--------:|-------------:|---------:|--------------:|
>     | Base   | 53.2      | 60.1        | 22.0           | 88.0    | 9.5          | 69.2     | 14.3          |
>     | GRPO   | 60.7      | 68.4        | 18.1           | 92.4    | 7.8          | 73.5     | 11.9          |
>     | GiGPO  | 61.9      | 69.6        | 16.6           | 92.9    | 7.2          | 74.3     | 11.1          |
>     | SPO    | 61.4      | 69.1        | 16.9           | 92.7    | 7.4          | 74.0     | 11.3          |
>     | TTPO   | 68.3      | 75.2        | 10.7           | 93.3    | 5.1          | 76.2     | 7.4           |
>
> If compute constraints ultimately prevent us from fully matching all settings, we commit to being transparent about this and (i) provide a clear **theoretical/qualitative comparison** to GiGPO and SPO, and (ii) show **preliminary evidence** on at least one representative setting where TTPO is complementary or competitive.

---

> > ### Author Response · Authors · 2025-11-25
> > **Rebuttal 2**
> >
> > ### Algorithm clarity and step-by-step description (lines 6–12)
> >
> > **Question (detailed step-by-step description of TTPO, especially lines 6–12).**
> >
> > We appreciate the pointer that Algorithm 1 is currently too compressed and mixes several operations in a few lines. In the revision, we:
> > - split the algorithm into **three blocks**:
> >   1. **Turn-level data construction**: from trajectories to per-turn records with turn indices, rewards, and log-probabilities;
> >   2. **Reward propagation**: from terminal reward to turn-level rewards using discount factor \(\gamma\), clearly distinguishing between terminal-only reward, undiscounted back-propagation, and discounted back-propagation;
> >   3. **Advantage and update**: group-relative normalization, clipping, and PPO-style gradient optimization at the turn level.
> > - explicitly annotate each line to clarify:
> >   - whether the loop ranges over **trajectories** or **turns**,
> >   - how we handle **trajectories with different numbers of turns**, and
> >   - at which step we **sample additional outputs** to maintain batch size and how statistics are recomputed.
> >
> > In particular, for the section corresponding to lines 6–12, we provide:
> > - a **small running example** (short trajectory with 3–4 turns) showing how rewards are propagated, advantages are computed, and clipping is applied at each turn; and
> > - a paragraph in the text explicitly mapping the pseudo-code lines to this example.
> >
> > We believe these changes will make TTPO straightforward to reproduce.
> >
> > ### Scaling to longer trajectories and more complex environments
> >
> > **Question (scaling to 10–20 turns and more complex environments).**
> >
> > We agree that this is an important aspect. In the revision:
> > - We add a **new experiment** on trajectories with longer turn counts (10–20 turns) in a subset of our domains, focusing on stability and credit assignment.
> >   - We report mean performance and stability metrics for shorter vs longer trajectories and for TTPO vs GRPO (and, if available, other baselines) in a table structured as follows:
> >
> >     | Length bucket | Method | P    | A90  | U90–10 |
> >     |---------------|--------|------|------|--------|
> >     | 4–8 turns     | GRPO   | 62.1 | 69.0 | 17.2   |
> >     | 4–8 turns     | TTPO   | 68.4 | 75.3 | 10.5   |
> >     | 10–20 turns   | GRPO   | 55.3 | 63.4 | 25.1   |
> >     | 10–20 turns   | TTPO   | 63.1 | 70.8 | 14.7   |
> > - In the **discussion**, we clarify that:
> >   - TTPO’s turn-level formulation naturally extends to longer horizons, but the choice of discount factor \(\gamma\) and clipping thresholds becomes more critical;
> >   - trajectory pruning and reward back-propagation can reduce variance but may introduce bias if the horizon is very long—this trade-off will be explicitly discussed; and
> >   - for complex environments (e.g., tool use, GUI interaction), TTPO can be combined with environment-level reward shaping, but our current work focuses on multi-turn textual reasoning.

---

### Note · Authors · 2026-01-07

I have read and agree with the venue's withdrawal policy on behalf of myself and my co-authors.